# Transaminase-catalysis to produce *trans*-4-substituted cyclohexane-1-amines including a key intermediate towards cariprazine
Emese Farkas [1,2,5] ✉, Péter Sátorhelyi[3], Zoltán Szakács [2], Miklós Dékány [2], Dorottya Vaskó[1], Gábor Hornyánszky [1], László Poppe [1,4] ✉ & János Éles [2] ✉

Cariprazine—the only single antipsychotic drug in the market which can handle all symptoms of bipolar I disorder—involves *trans*-4-substituted cyclohexane-1-amine as a key structural element. In this work, production of *trans*-4-substituted cyclohexane-1-amines was investigated applying transaminases either in diastereotope selective amination starting from the corresponding ketone or in diastereomer selective deamination of their diastereomeric mixtures. Transaminases were identified enabling the conversion of the *cis*-diastereomer of four selected *cis/trans*-amines with different 4-substituents to the corresponding ketones. In the continuous-flow experiments aiming the *cis* diastereomer conversion to ketone, highly diastereopure *trans*-amine could be produced (*de* > 99%). The yield of pure *trans*-isomers exceeding their original amount in the starting mixture could be explained by dynamic isomerization through ketone intermediates. The single transaminase-catalyzed process—exploiting the *cis*-diastereomer selectivity of the deamination and thermodynamic control favoring the *trans*-amines due to reversibility of the steps—allows enhancement of the productivity of industrial cariprazine synthesis.

The efficient stereoselective synthesis of amines is one of the supreme wishes of small molecule drug discovery and development due to their relevance in medicinal chemistry—as evidenced by statistics[1,2]. Numerous industrial scale endeavors (e.g., ChiPros™ process with lipase[3–8], or manufacturing of sitagliptin with transaminase[9] utilizing emerging technologies[10] and key green engineering research areas[11] like biocatalysis[12–19] and flow chemistry[20–23] can slake this thirst in a sustainable way. Enzyme-catalyzed deracemization strategies[24–36] with full conversion of the racemate proved to be a viable strategy for the synthesis of pure enantiomers of amines. In contrast, enzyme-catalyzed dynamic de-diastereomerizations are quite rare[37].

Cariprazine, developed by Gedeon Richter Plc. and being marketed as Reagila® in Europe and as Vraylar® in the USA, (Fig. 1a) is an atypical antipsychotic (third generation of neuroleptics) for the treatment of schizophrenia, major depression disorder, and mania, mixed as well as

depression episodes associated with bipolar I. disorder[38–43]. A substantial structural constituent of cariprazine is the bitopic cyclohexyl moiety *trans*-**1a**—containing two centers of pseudoasymmetry, 1*r* and 4*r* providing the *trans* arrangement—which connects the two pharmacophores (Fig. 1a). The key intermediate is synthesized by a two-step hydrogenation of 4-nitrophenylacetic acid followed by an ethyl ester formation (Fig. 1b)[44]. Separation of the diastereomeric mixture *cis/trans*-**1a** with classical crystallization affords the key intermediate *trans*-**1a** in moderate yield governed by the starting *cis:trans* ratio in extremely high diastereomeric purity[44]. Due to the non-acidic nature of hydrogens attached to the pseudoasymmetric centers, chemical isomerization of *cis*-**1a** is extremely difficult.

The productivity of this patented industrial process can be improved by swapping from crystallization either to a diastereotope selective production of *trans*-**1a** (Fig. 1c, for nomenclature of types of stereoselectivities, see p. 129 in ref. 45, or to a diastereomer selective amine-to-ketone conversion of the

[1]Department of Organic Chemistry and Technology, Budapest University of Technology and Economics, Műegyetem rkp. 3, 1111 Budapest, Hungary. [2]Gedeon Richter Plc., PO Box 27, 1475 Budapest, Hungary. [3]Fermentia Microbiological Ltd., Berlini út 47-49, 1405 Budapest, Hungary. [4]Biocatalysis and Biotransformation Research Centre, Faculty of Chemistry and Chemical Engineering, Babeș-Bolyai University of Cluj-Napoca, Arany János str. 11., 400028 Cluj-Napoca, Romania. [5]Present address: Gedeon Richter Plc., PO Box 27, 1475 Budapest, Hungary. ✉e-mail: farkaseme@richter.hu; poppe.laszlo@vbk.bme.hu; j.eles@richter.hu

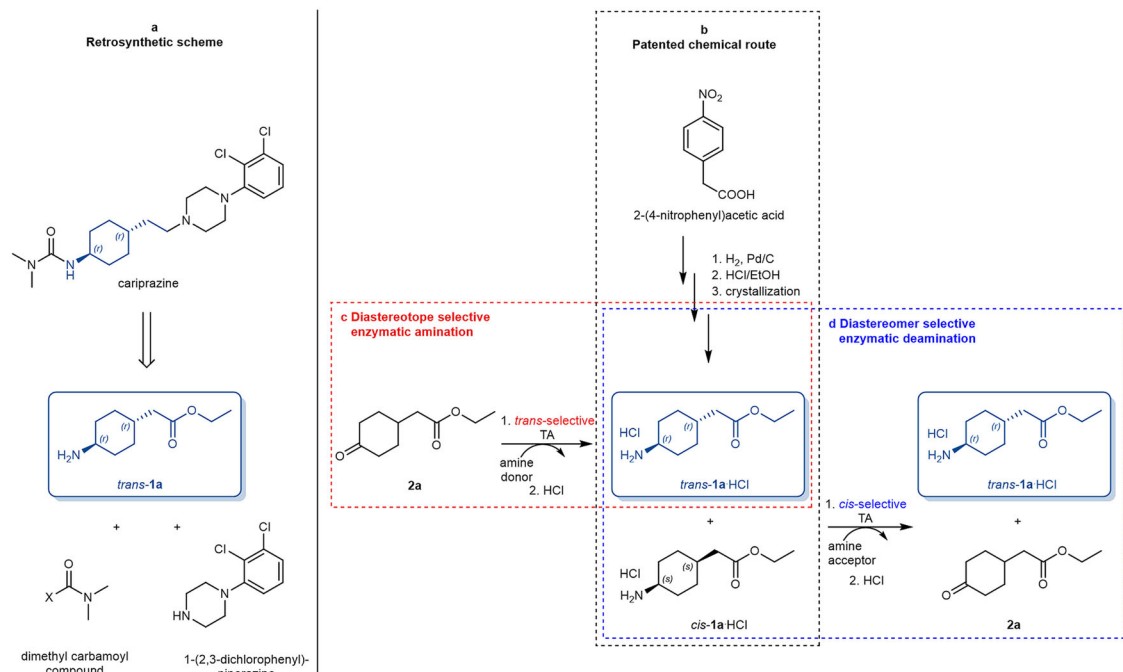

**Fig. 1 | The synthesis of the key intermediate of cariprazine. a** Retrosynthetic scheme for cariprazine. **b** Patented chemical route to the key intermediate (box of black dotted line). Transaminase-catalyzed alternative synthetic routes for cariprazine key intermediate *trans*-**1a** can exploit **c** proper diastereotope selectivity of transaminase-catalyzed amination of the corresponding ketone **2a** (box of red dotted line) or **d** the proper diastereomer selectivity of a transaminase preferring deamination of the *cis*-diastereomer *cis*-**2a** from the *cis/trans*-diastereomeric mixture forming in the chemical route (box of blue dotted line).

*cis*-**1a** (Fig. 1d) applying proper biocatalysts followed by removal of the non-protonable ketone **2a**. Although several biocatalysts—such as transaminase (TA)[46–48], lipase[49–51], or ketoreductase[52,53]—were applied for stereoselective biotransformations of cyclohexane-1-amines with substituents at 2- or 3-position, the presence of stable centers of asymmetry rendered these processes significantly dissimilar to our case with the 4-substituted derivatives having only centers of pseudoasymmetry. The closest analogy to our case is the application of reductive aminase for the production of isopropyl *cis*-3-aminocyclobutane-1-carboxylate—the intermediate of abrocitinib—having also two centers of pseudoasymmetry[54]. However, no biocatalytic processes have been developed for preparing either the diastereomer of **1a** or even the 4-substituted cyclohexane-1-amine skeleton in a stereoselective way until so far.

Here, we report a process catalyzed by a single transaminase to produce *trans*-4-substituted cyclohexane-1-amines *trans*-**1a-d** from the corresponding *cis/trans*-diastereomeric mixtures. During the diastereomer selective *cis*-deamination approach in continuous-flow mode aiming *trans* 4-substituted cyclohexane-1-amines *trans*-**1a-d** with high diastereomeric excess, a dynamic *cis*-to-*trans* isomerization was revealed—being potentially applicable to enhance the productivity of industrial cariprazine synthesis[55].

## Results and discussion

First, the diastereotope selective amination aided by various TAs has been investigated with four 4-substituted cyclohexanones **2a-d** representing various properties (Fig. 2). Convenient synthetic routes have been developed for **2a-d** and for the diastereomeric mixtures of amines **1a-d** (for details, see Section 3.1.1 and 3.1.2 in Supplementary information, respectively)[55,56].

For amination of the ketones **2a-d** in preliminary enzyme screening, six transaminases involving three (*R*)-selective—the TA from *Arthrobacter sp.* (*ArR*-TA)[57], its mutated variant (*ArR*-TA$_{mut}$)[46], from *Aspergillus terreus* (*AtR*-TA)[58]—and three (*S*)-selective—the TA from *Arthrobacter citreus* (*ArS*-TA)[59], from *Vibrio fluvialis* (*VfS*-TA)[60], and mutant of TA from *Chromobacterium violaceum* (*CvS*-TA$_{W60C}$)[61,62]—in immobilized whole-cell forms[63] were selected using the corresponding enantiomer of 1-

phenylethan-1-amine (*R*)-**3** or (*S*)-**3** as amine donor at 30 °C in batch mode (Fig. 2). For these experiments, recombinant *E. coli* whole-cells over-expressing one of the above TAs were immobilized together with hollow silica microspheres as support by entrapment in a sol-gel system. Details on the expression of *ArR*-TA, *ArR*-TA$_{mut}$, *AtR*-TA, *ArS*-TA, *VfS*-TA, and *CvS*-TA$_{W60C}$ and immobilization of TA-expressing whole-cells were published in our preceding work[63]. Further details can be found in Supplementary Information Section 1.3.2. The sol-gel entrapment method combined the advantages of cell-adsorption on silica microspheres providing good mechanical properties of the biocatalyst with high immobilization yield (~100% of the cells were retained; ~0.9 g of dry TA biocatalyst could be produced from 1 g of wet cells) by the entrapping silica matrix. The immobilization could be scaled up from a g scale to 10 g scale without any noticeable problem[63]. To explore the reaction parameters (heat and cosol-vent tolerance of imm-*CvS*-TA$_{W60C}$), the kinetic resolution of racemic 4-phenylbutane-2-amine *rac*-**8** was applied as a test reaction (Sections 1.3.4.1–1.3.4.3 of Supplementary Methods, and Supplementary Figs. S7–S13) with slight modifications compared to our previous studies[63].

### Preliminary screening of diastereotope selective amination of ketones 2a-d with immobilized whole-cell forms transaminases in batch mode

Ketones **2a-d**—selected as model substrates for this study—were investigated for amination using six transaminases in their immobilized whole-cell form (Fig. 2). After the synthesis and structural characterization of pure *trans*-**1a-d**·HCl (for details, see Section 3.1 in Supplementary Information), the diastereotope selectivity in the batch mode reactions could be analyzed by GC.

Irrespective of their established enantiomer preferences, the formation of *cis*-diastereomer of the amines **1a-d** was favored with almost all investigated TAs. Interestingly, *trans*-diastereomer preference was observed only in the case of **1c** with *ArR*-TA$_{mut}$. High conversions from the ketones **2a-d** could be achieved with *ArR*-TA$_{mut}$, *CvS*-TA$_{W60C}$, and *VfS*-TA providing high diastereomeric excess with only the latter two TAs. *ArR*-TA$_{mut}$—despite its superior performance on enantiotope selective amination of

**Fig. 2 | Preliminary screening for diastereotope selective amination of the ketone 2a-d with immobilized whole-cell forms of three (R)-selective and three (S)-selective transaminases in batch mode.** Reaction conditions: 2a-d (5 mM), (R)-3 (15 mM, 3 eq., for ArR-TA, ArR-TA$_{mut}$, AtR-TA), (S)-3 (15 mM, 3 eq., for ArS-TA, VfS-TA, CvS-TA$_{W60C}$), PLP (0.05 mM, 5 mol%) and immobilized whole-cell transaminase (imm-whc-TA, 50 mg) in sodium phosphate buffer (100 mM, pH 7.5) shaking at 600 rpm for 24 h at 30 °C. Conversion (left scale of the Y-axis) and diastereomeric excess (de: right scale of the Y-axis) were determined by GC.

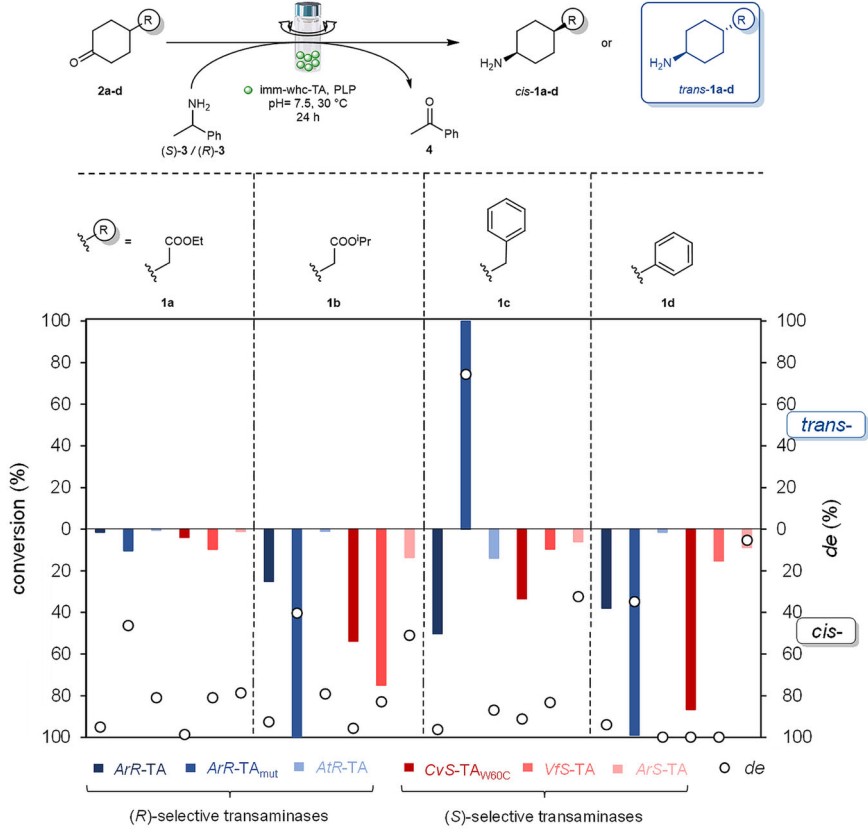

bulky ketones[9]—provided only low or moderate diastereomeric ratio. Relatively high diastereomeric excesses of the cis-amines cis-1a-d could be reached by VfS-TA and CvS-TA$_{W60C}$. The cis-diastereomeric preference of AtR-TA was also high, but the conversions of the ketones 2a-d under these operation conditions were unsatisfactory (<8%). While aminations of 2b-d with ArR-TA with cis-diastereomeric preference (de > 92%) proceeded in moderate conversions, the conversion from ketone 2a remained low (<2%). The phenyl-bearing ketone 2d was investigated with CvS-TA, CvS-TA$_{W60C}$, and in aminations performed in organic solvents indicating similar cis-stereopreference as our results in aqueous systems[64].

Since we could not find any TA capable of catalyzing the diastereotope selective amination with the desired trans-diastereomeric preference to provide the targeted amine diastereomers trans-1a-d (Fig. 1c), our attention turned to the diastereomer selective deamination which was enabled by the cis-preference (Fig. 1d). An expected benefit of the deamination strategy from the cis/trans-1a-d mixtures was the possibility of separation of the unreacted trans-1a-d from the corresponding ketone 2a-d without the need of recrystallization.

**Deamination of cis/trans-1a-d by immobilized transaminases in continuous-flow mode**

The isomer separation strategy (Fig. 1d) based on the diastereomer selective deamination of the cis-amines cis-1a-d was performed with VfS-TA and CvS-TA$_{W60C}$ mostly in batch mode using different forms—native soluble enzyme, immobilized whole-cells, and purified form immobilized on polymer resin[65]. The efficiency of biotransformations can be significantly improved by combining enzyme immobilization techniques and continuous-flow reactor systems[66]. Since CvS-TA$_{W60C}$ could be efficiently immobilized on polymer resin[65], we continued our experiments with this form of the transaminase in continuous-flow mode for obtaining trans-1a-d, inter alia, the pharmaceutical important trans-1a on a preparative scale.

The CvS-TA$_{W60C}$ (purified by standard Ni-NTA method) was immobilized on bisepoxide-activated macroporous aminoalkyl resins. Previous experiments with pure CvS-TA$_{W60C}$ (overexpressed in E. coli and

purified by standard Ni-NTA method, Supplementary Fig. S6) showed that the best resin-immobilized TA form with high operational stability could be obtained by covalent attachment of the enzyme to glycerol diglycidyl ether activated ethylamine-functionalized mesoporous polymer (for details, see Section 1.3.3 in Supplementary Information)[65]. Since this immobilized form of CvS-TA$_{W60C}$ (imm-CvS-TA$_{W60C}$) proved to be suitable for enantiomer separation of various amines in continuous-flow mode using sodium pyruvate as amine acceptor[65], a similar approach for the diastereomer separation of cis/trans-1a-d seemed to be straightforward by deamination of cis-diastereomer from the mixture to leave the desired unreacted trans-1a-d and the corresponding ketone 2a-d (Fig. 3a). Because in the kinetic resolution of racemic amines employing a stoichiometric equivalent of the amino acceptor—usually pyruvate (or related α-keto acid)—the thermodynamic equilibrium is on the product side[67], the diastereomer selective approach seemed to be favored as well.

Because the imm-CvS-TA$_{W60C}$ was investigated in continuous-flow mode in kinetic resolutions of racemic amines only at ambient temperature (30 °C) for short reaction times (~1 h)[65], investigations for extending the reaction conditions were performed first (for details, see Section 1.3.4 in Supplementary Information). Based on these results, deamination in continuous-flow mode was accomplished at 40 °C providing significantly enhanced reaction rates with sufficiently high operational stability. In the continuous-flow system (Supplementary Fig. S14), the buffered solution (pH 7, HEPES) supplemented with the diastereomeric mixtures of the amines cis/trans-1a-d·HCl, sodium pyruvate, dimethyl sulfoxide (DMSO) (to enhance substrate solubility) and PLP was pumped through the packed-bed reactor filled with immobilized CvS-TA$_{W60C}$. Before starting the deamination by applying substrate-containing input stream, the imm-CvS-TA$_{W60C}$-filled columns were conditioned with the flow of buffer solution containing PLP (Fig. 3a, with ~5×volume of the void of the actual system). The preliminary continuous-flow experiments were started in single-column systems at 40 °C using a flow rate of 10 μL min$^{-1}$. Perfect diastereomeric excess (de > 99.5%) in a single-column could be achieved only in diastereomer selective deamination of cis/trans-1c (Entry 3 in Table 1). For

**Fig. 3 | Diastereomer selective deamination from the *cis/trans*-1a-d mixtures catalyzed by immobilized *CvS*-TA$_{W60C}$ in continuous-flow mode.**
**a** Scheme of biocatalytic deamination from the *cis/trans*-**1a-d** mixtures in a continuous-flow reactor system consisting of immobilized *CvS*-TA$_{W60C}$ filled packed-bed reactor(s). **b** Time course of the diastereomer selective deamination from the *cis/trans*-**1a-d** mixtures in the continuously operated packed-bed reactor system (the molar fractions of *trans*-**1a-d**—also considering the forming ketone **2a-d**—in the effluent are shown as a function of operational time).

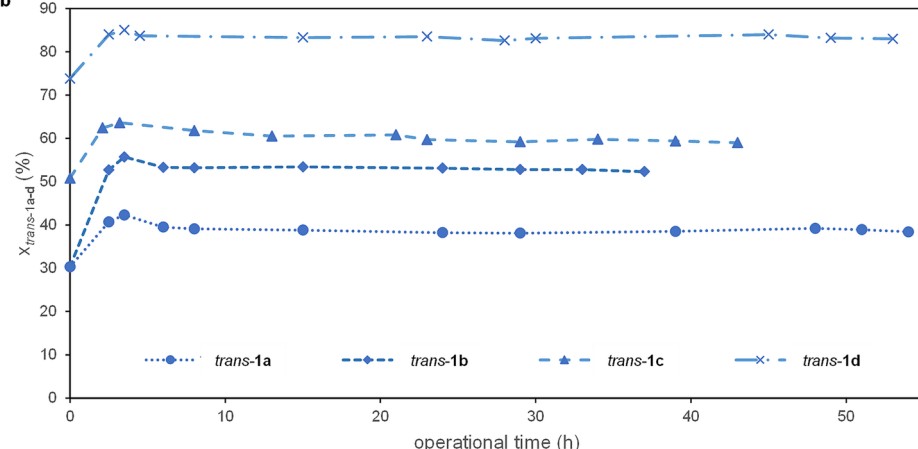

<table>
<tr><th></th><th></th></tr>
</table>

## Table 1 | Diastereomer selective deamination of *cis/trans*-1a-d mixture catalyzed by immobilized *CvS*-TA$_{W60C}$ in continuous-flow mode

| Entry | Amine | $c_{1a\text{-}d}$[a] (mM) | $n$[b] (-) | $x_{cis}/x_{trans}$[c,d] (-) | $x_{2a\text{-}d}$[d,e] (%) | $x_{trans}$[d,e] (%) | $de_{trans}$[d,f] (%) | Yield[g] (%) | STY[h] (kg m$^{-3}$ d$^{-1}$) | $r_{flow}$[i] (µmol g$^{-1}$ min$^{-1}$) | $\tau_{app}$[j] (min) |
|---|---|---|---|---|---|---|---|---|---|---|---|
| 1 | **1a** | 20 | 3 | 69.7/30.3 | 61 | 39 | >99.5 | 31 | 8.0 | 0.055 | 162 |
| 2 | **1b** | 20 | 4 | 51.9/48.1 | 50 | 50 | >99.5 | 47 | 9.7 | 0.062 | 216 |
| 3 | **1c** | 15 | 1 | 49.2/50.8 | 40 | 59 | >99.5 | 54 | 30.3 | 0.203 | 54 |
| 4 | **1d** | 15 | 2 | 26.2/73.8 | 17 | 81 | >99.5 | 78 | 23.2 | 0.166 | 108 |

[a]Reaction conditions: *cis/trans*-**1a-d** as HCl salt, sodium pyruvate (1 equiv. to *cis/trans*-**1a-d**) and PLP cofactor (1 n/n%) in HEPES (50 mM, pH = 7.0) with DMSO (10 v/v% for **1a,b**; 20 v/v% for **1c,d**) in serially coupled (*n*) packed-bed reactors at 40 °C with 10 µL min$^{-1}$. For details, see Methods.
[b]Number of column(s) filled with imm-*CvS*-TA$_{W60C}$ (filling mass: 375±12 mg/column). Further data for column number selection are shown in Supplementary Table S2.
[c]$x_{cis}/x_{trans}$ = (molar fraction of *cis*-**1a-d**)/(molar fraction of *trans*-**1a-d**) in the starting reaction mixture (influenced by the chemical reductive amination from the starting ketones **2a-d**).
[d]Determined after derivatization with acetic anhydride by GC, for details, see Supplementary Data 2.
[e]Molar fractions (*trans*-**1a-d**, also considering the forming ketone **2a-d**) in the effluent during stationary state.
[f]Diastereomeric excess of *trans*-**1a-d** in the effluent during stationary.
[g]Isolated yield of *trans*-**1a-d** (from stationary state of the reactions, see Methods).
[h]Space-time yield (*STY*) at stationary state; determined by using the *STY* = $m_{1a\text{-}d}$ / ($V_{react} \times t_{coll}$) equation, where $m_{1a\text{-}d}$ is the mass of isolated *trans*-**1a-d** (kg), $V_{react}$ is the total volume of the packed-bed column reactor (m³), and $t_{coll}$ is the time of effluent collection (d).
[i]Specific reaction rate ($r_{flow}$) at stationary state; determined by using the $r_{flow}$ = $n_{1a\text{-}d}$ / ($m_b \times t_{coll}$) equation, where $n_{1a\text{-}d}$ is the molar amount of isolated *trans*-**1a-d** (µmol), $m_b$ is the total mass of the biocatalyst in the reactor (g), and $t_{coll}$ is the time of effluent collection (min).
[j]Apparent residence time ($\tau_{app}$); determined by using the $\tau_{app}$ = $V_{react}$ / $f$ equation, where $V_{react}$ is the total volume of the serial packed-bed column reactor (mL), and $f$ is the flow rate (mL min$^{-1}$), see Section 1.3.4.4.2 in Supplementary Information).

amines *cis/trans*-**1a,b,d**, where the diastereomeric excess of the residual *trans*-amine remained lower than 99%, the number of columns was increased (Entries 1,2,4 of Table 1, see Section 1.3.5.1 in Supplementary Information).

The long-term operational stability of immobilized *CvS*-TA$_{W60C}$ biocatalyst was apparent during the preparative scale continuous-flow operations of diastereomer selective deamination starting from *cis/trans*-**1a-d**·HCl at 40 °C in HEPES buffer containing DMSO (Table 1 and Section 1.3.5 in Supplementary Information). The system could be operated for at least 36 h (including at least 24 h stationary operation) without observable loss of activity for each diastereomeric mixture *cis/trans*-**1a-d** (Fig. 3). According to our preliminary investigations (Section 1.3.4.4.3 in Supplementary information and Supplementary Fig. S15) the immobilized *CvS*-TA$_{W60C}$ biocatalyst-filled columns lost a part of their activity after 24 h stationary operation, with ~40–50% residual activity in the 36–70 h period.

Assuming high diastereomer selectivity in an irreversible process, the expected amount of the residual *trans*-**1a-d** could not exceed the original amount in the starting diastereomeric mixture *cis/trans*-**1a-d**. Entries 1,3,4 of Table 1, however, indicated significantly less amount of the forming ketone (**2a,c,d**, respectively) as compared to the molar fraction of the *cis*-amine in the starting *cis/trans*-**1a,c,d** mixture. The significantly higher isolated yield of the diastereopure *trans*-amine (*trans*-**1c,d**, Entries 3,4 of Table 1) than the original amount in the diastereomeric mixture further confirmed these results.

Although the space-time yield (*STY*) of the non-optimized production of *trans*-**1a-d** (8–30 kg m$^{-3}$ d$^{-1}$ = 0.33–1.26 × 10$^{-3}$ kg·L$^{-1}$·h$^{-1}$, see Table 1) remained beyond the >1 kg L$^{-1}$ h$^{-1}$ values required for an industrially viable process[68], the *STY* values reported here can be considered as acceptable values at laboratory scale. However, compared to the 103 kg m$^{-3}$ d$^{-1}$ *STY* value achieved in our previous on lipase-catalyzed dynamic-kinetic-resolution of benzylic amines in continuous-flow PBR system[36], the *STY* of the transaminase-catalyzed process reaching 7–20% of the lipase-catalyzed method is promising. The lower STY's can be rationalized by taking specific reaction rates ($r_{flow}$) into account. While lipases known as the workhorses of

**Fig. 4 | The dynamic isomerization of *cis*- to *trans*-1a-d catalyzed by a single transaminase.** The thermodynamic equilibrium shifts the process towards *trans*-isomer formation. (The balance and brain motifs of the Figure were accessible under Creative Commons CC0 by SVG Silh (svgsilh.com) and by PublicDomainPictures (www.publicdomainpictures.net), respectively).

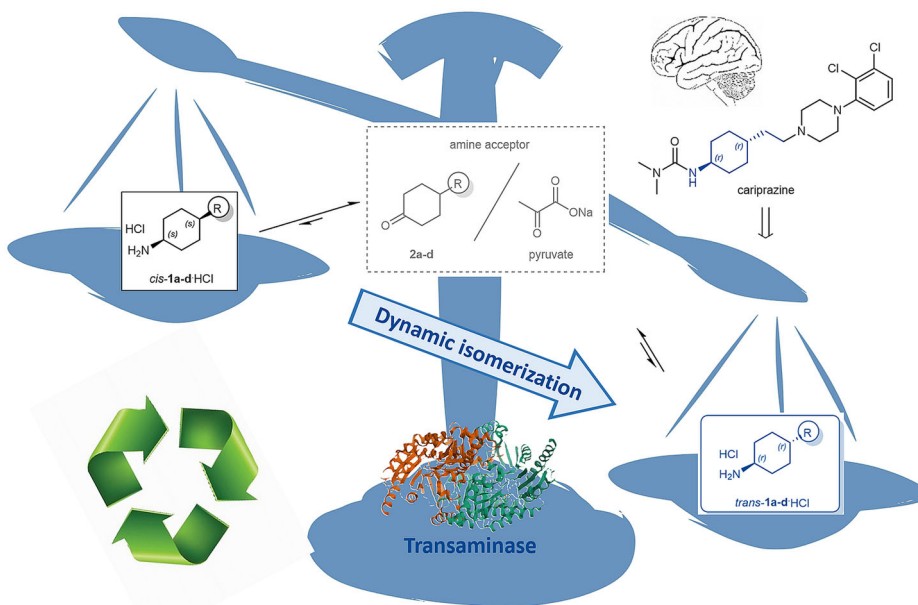

biocatalysis even at an industrial scale can provide $r_{flow}$ values well above 10–100 µmol g$^{-1}$ min$^{-1}$ [66], the transaminase imm-*CvS*-TA$_{W60C}$ being not engineered for this purpose afford only $r_{flow}$ = 0.055–0.203 µmol g$^{-1}$ min$^{-1}$ for production of *trans*-**1a-d** (Table 1). The apparent residence times ($\tau_{app}$ = 54–216 min) estimated for the production of *trans*-**1a-d** (Table 1) were in good agreement with the times required to reach the stationary state (2–5 h) in the continuous-flow systems (Fig. 3).

The greater-than-expected amount of *trans*-**1a-d** could be rationalized by assuming a dynamic isomerization catalyzed by a single transaminase (imm-*CvS*-TA$_{W60C}$). Since the TA-catalyzed processes are reversible, the diastereomer selective deamination process is not fully irreversible and the partial reversibility allows the prevalence of thermodynamic consequences. Since the *trans*-diastereomers are thermodynamically more favored, a process exploiting the combination of kinetic selectivity and thermodynamic equilibrium can surpass the limitation of the *trans*-diastereomer amount in the original mixture by the dynamic isomerization.

Although TAs[46–48] were applied for stereoselective biotransformations of cyclohexane-1-amines with substituents at 2- or 3-position, the presence of two stable centers of asymmetry in such compounds renders their diastereomer selective transformations significantly different from the case of the 4-substituted derivatives. The stereochemistry within the 4-substituted cyclohexane-1-amines **1a-d** is determined by two centers of pseudoasymmetry at the 1- and 4-substituents at the cyclohexane ring (1*r*,4*r* for *trans*, or 1*s*,4*s* for *cis*; the centers of pseudoasymmetry are marked *r/s* in the CIP system, see pp. 52–54 in ref. 45). A unique feature of such systems is that if one center of pseudoasymmetry is eliminated (e.g., an 1*r* center is destroyed upon deamination to ketone **2a-d**), the other (in this case 4*r*) is also eradicated without altering any of the four covalent bonds directly attached to the central atom. In this respect, the *cis/trans*-stereochemistry is defined by the two centers of pseudoasymmetry behaving as a single stereogenic unit.

Importantly, the *cis*-to-*trans* dynamic isomerization observed during the diastereomer selective deamination of *cis/trans*-**1a-d** mixture in continuous-flow mode could be catalyzed by a single transaminase—by imm-*CvS*-TA$_{W60C}$ in this study. The favored formation of *trans*-**1a-d** is based on the *cis*-diastereomer selectivity in the deamination of the *cis/trans*-**1a-d** mixture and the moderate diastereotope selectivity in the amination of the corresponding ketones **2a-d**, since the reversibility of the process allows thermodynamic control leading to the more stable *trans*-diastereomers *trans*-**1a-d**. The fact that using 1 molar equivalent pyruvate (100% of amine acceptor) resulted in only 17–60% ketone formation could be due to a mixture of kinetic and thermodynamic reasons. Thus, in the process of using TA in a packed-bed reactor under continuous-flow conditions both

kinetic and thermodynamic components play a role. Due to the large amount of catalyst, kinetic selectivity is more pronounced, and almost diastereomerically pure *trans*-amines are forming but partial equilibration (thermodynamics) allows isomerization to the more stable isomer and thereby formation of *trans*-amines in higher amounts than present in the original mixture (Fig. 4). Usually, increasing pyruvate concentration increases the rate of *cis*-deamination (mostly influenced by kinetic factors). On the other hand, increasing pyruvate concentration increases the proportion of the 1-cyclohexanone-compound in the final mixture (mostly influenced by thermodynamic factors)[64].

The requirement of only a single enzyme for the dynamic *cis*-to-*trans*-isomerization of 4-substituted cyclohexane-1-amines **1a-d** is in stark contrast to the deracemization of the racemic amines which is also a dynamic process based on enantiomer selectivity and reversibility of enzyme catalysis. Due to the same energy of the two enantiomers in deracemization, the governing effect of thermodynamic preference cannot be exploited and the process can only be performed with two enantiocomplementary enzymes (sequential usage or remarkably different kinetics profiles)[24,25].

Consequently, a single transaminase—such as *CvS*-TA$_{W60C}$ with the proper diastereomer selectivity—could be applied for interconversion of the *cis*- to the *trans*-diastereomer of 4-substituted cyclohexane-1-amines **1a-d**, a process being useful for enhancing the industrial process of the cariprazine synthesis (Fig. 4)[64]. The patent application on using this dynamic isomerization process for enhancing the cariprazine synthesis involves further examples —with proper single transaminases such as *CvS*-TA, *VfS*-TA or their various mutants—achieving up to 80% *cis*-to-*trans* conversion from the *cis*-**1a** isomer providing *trans*-**1a** in ~80% *de* nearby the thermodynamic equilibrium[64].

## Conclusion

In conclusion, transaminases with *cis*-selectivity could be applied to produce the *trans* 4-substituted cyclohexane-1-amines from the corresponding *cis/trans* diastereomeric mixtures by selective deamination. Implementing the process in continuous-flow mode using packed-bed reactors filled with immobilized W60C mutant of the transaminase from *Chromobacterium violaceum* could produce the targeted *trans*-4-substituted cyclohexane-1-amines in *de* > 99%. The yield of pure *trans*-isomers exceeding their original amount in the starting mixture revealed a dynamic isomerization via the ketone intermediate. Although dynamic equilibration-based processes were already exploited to shift the composition of enantiomeric mixtures of amines (i.e., racemic amines) towards the direction of either enantiomers, these so-called deracemizations could exploit only kinetic differentiation and—due to the thermodynamic equivalency of the enantiomers—required

two transaminases with different kinetics and enantiopreference. In contrast, the dynamic isomerization process converting *cis/trans* diastereomeric mixtures of 4-substituted cyclohexane-1-amines to the thermodynamically favored *trans* diastereomer required only a single transaminase with proper diastereomer preference. The results with four different 4-substituted cyclohexane-1-amines—in which two centers of pseudoasymmetry determining the *trans* or *cis* stereochemistry behave as one stereogenic unit—requiring only a single transaminase to convert the thermodynamically less favored *cis*- to the more favored *trans*-diastereomer. Besides the general applicability of this type of dynamic isomerization, this bioprocess provides an appealing approach to improve the industrial synthesis of cariprazine by converting the *cis/trans*-diastereomeric mixture to *trans*-ethyl 2-(4-aminocyclohexyl)acetate—a key intermediate of the process.

## Methods
### General
GC-FID analysis was performed on Agilent 5890 GC equipped with a nonpolar HP-5 column (Agilent J&W; 30 m × 0.25 mm × 0.25 µm film thickness of (5%-Phenyl)-methylpolysiloxane). High-resolution Mass Spectrometry (HR-MS) and coupled Mass Spectrometry-Mass Spectrometry (MS-MS) analyses were performed on a Thermo Velos Pro Orbitrap Elite (Thermo Fisher Scientific) system. 1D and 2D NMR spectra were recorded on a Avance III HDX spectrometers from Bruker BioSpin GmbH, Rheinstetten, Germany. Infrared spectra were recorded on a Bruker ALPHA FT-IR spectrometer and wavenumbers of bands are listed in $cm^{-1}$. Column chromatography was performed on silica gel (Kieselgel 60, 0.040–0.063 mm, 230–400 mesh, Merck, Darmstadt, Germany). TLC was carried out using Kieselgel 60 F254 (Merck) sheets. Commercially available chemicals and reagents were purchased from Sigma Aldrich (Saint Louis, MO, USA), Alfa Aesar Europe (Karlsruhe, Germany), Merck (Darmstadt, Germany), Fluka (Milwaukee, WI, USA), Materium Innovations (Granby, QC, Canada), Resindion S.r.l. (Binasco, Italy). Transaminases (*ArR*-TA, *ArR*-TA$_{mut}$, *AtR*-TA), *ArS*-TA, *VfS*-TA, and *CvS*-TA$_{W60C}$) were immobilized in whole-cell form (imm-whc-TAs)[62] or in porous polymeric beads by covalent binding (imm-*CvS*-TA$_{W60C}$)[65]. Details on enzyme coding plasmids, transaminase sequences, and immobilization methods are described in Section 1.2, Section 3.2, and 3.3. in Supplementary information; respectively. CatCart™ columns (inner diameter: 4 mm; total length: 70 mm; packed length: 65 mm; inner volume: 0.816 mL) were products of ThalesNano Plc. (Budapest, Hungary).

### Synthesis of ketones 2a-d and the diastereomeric mixtures of the amines *cis/trans*-1a-d
The preparation of the ketones is shown in Supplementary Fig. S1 and described in Section 1.3.1.1. of Supplementary Methods.

### Amination of ketones 2a-d with immobilized whole-cell TAs in batch mode in an analytical scale
The immobilized whole-cell TA biocatalyst (50 mg) was suspended in phosphate buffer (100 mM, pH 7.5) containing pyridoxal-5'-phosphate monohydrate (PLP, 0.05 mM, 5 n/n%) in 4 mL vials and incubated for 1 h. Then the corresponding enantiomer of 1-phenylethan-1-amine [15 mM, 3 eq.; (*S*)-**3** for *ArS*-TA, *VfS*-TA, *CvS*-TA$_{W60C}$ and (*R*)-**3** for *ArR*-TA, *ArR*-TA$_{mut}$, *AtR*-TA] was added to the suspension and incubated for 30 min again. Finally, the ketones (**2a-d**, 5 mM) were added. The final reaction volume was 1 mL. The reaction mixture was shaken on an orbital shaker (600 rpm) at 30 °C for 24 h. After the addition of sodium hydroxide (100 µL, 1 M) to the samples taken from the reaction mixture (800 µL), the resulting mixture was extracted with ethyl acetate (800 µL). After derivatizing the amines **1a-d** in the extract by addition of acetic anhydride (10 µL, 60 °C, 1 h), the organic phase was dried over Na$_2$SO$_4$ and analyzed by gas chromatography [on Agilent 5890 equipment, FID detector and HP-5 column (Agilent J&W; 30 m × 0.25 mm × 0.25 µm film thickness of (5%-phenyl) methylpolysiloxane); H$_2$ carrier gas, injector: 250 °C, detector: 250 °C, head pressure: 12 psi, split ratio: 50:1]. GC methods and reference chromatograms of ketones **2a-d** and diastereomeric acetamides *cis/trans*-**7a-d** are

shown in Supplementary Table 1, and Supplementary Figs. S2-S5, respectively. Further GC data are provided in Supplementary Data 2 as Figure GC1-GC4.

### Deamination of *cis/trans*-1a-d with immobilized *CvS*-TA$_{W60C}$ in continuous-flow mode
After filling CatCart™ columns with covalently immobilized *CvS*-TA$_{W60C}$ biocatalyst (filling weight = 375±12 mg/column), the system (detailed in Section 1.3.4.4 of Supplementary Information) consisting of the indicated number of *CvS*$_{W60C}$-TA-filled columns (see Table 1) was prewashed by HEPES buffer (50 mM, pH = 7.0) containing PLP (~5× volume of the column void). Then the substrate solution [**1a-b** (20 mM) or **1c-d** (15 mM) in HEPES buffer (50 mM, pH = 7.0) containing DMSO as cosolvent (10% v/v for **1a-b**, 20% v/v for **1c-d**), sodium pyruvate (1. eq.) and PLP (1% n/n)] was pumped through the column thermostated at 40 °C with 10 µL min$^{-1}$ of flow rate. Samples were taken from the effluent and analyzed by GC (as described in Section 1.3.1.3 of Supplementary Information). After the stationary operation was established (5–8 h from the start of the experiment, depending on the number of *CvS*$_{W60C}$-TA-filled columns), the product mixture was collected for 30–48 h.

From the mixture collected during the stationary phase of the operation, the pure *trans*-product *trans*-**1a-d** was isolated. The pH of the mixture was set to 1 by the addition of aqueous cc. HCl, followed by removal of the formed ketone **2a-d** by extraction with dichloromethane (3 × 50 mL). Then the pH of the aqueous phase was set to 12 by the addition of ammonium hydroxide (25%v/v) and *trans*-**1a-d** was extracted with dichloromethane (3 × 50 mL). The unified organic phase was washed with brine (30 mL) and dried over Na$_2$SO$_4$. After the removal of the solvent in a vacuum, the residue was dissolved in diethyl ether (5 mL). The precipitate forming by HCl-gas treatment was collected by filtration and dried to give the corresponding *trans*-amine hydrochloride salt (*trans*-**1a-d**·HCl). NMR data are provided in Section 1.3.5.2 in Supplementary Information and in Supplementary Data 1.

### Reporting summary
Further information on research design is available in the Nature Portfolio Reporting Summary linked to this article.

## Data availability
All data generated or analyzed during this study are included in this published article (and its supplementary information files). Additional data that support the findings of this study i.e., supplementary methods, detailed experimental procedures and characterizations of new compounds, synthetic procedures for substrates, details of the equipment for continuous-flow reactions, physicochemical data of the products, as well as analytical separation conditions of the studied compounds by GC, are available in the Supplementary Information. Supplementary Methods, Notes, Tables, and Figures are available in the Supplementary Information. NMR spectra of all non-commercial ketones **2a-c**, HCl salt of diastereomeric amine mixtures *cis/trans*-**1a-d**·HCl, and HCl salt of the produced pure *trans*-amines *trans*-**1a-d**·HCl are available in the Supplementary Data 1. GC chromatograms for transaminase-catalyzed aminations and deaminations are presented in the Supplementary Data 2. IR spectra of all non-commercial ketones **2a-c**, HCl salt of diastereomeric amine mixtures *cis/trans*-**1a-d**·HCl are available in Supplementary Data 3. Supplementary Data 4–7. contain primary data for figures and tables in the manuscript.

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

## Acknowledgements

The research reported in this paper is part of project no. TKP2021-EGA-02, implemented with the support provided by the Ministry for Innovation and Technology of Hungary from the National Research, Development and Innovation Fund (NRDIF), financed under the TKP2021 funding scheme. Funding from NRDIF to improve protein production infrastructure (NKP-2018-1.2.1-NKP-2018-00005) is acknowledged.

## Author contributions

E.F., L.P., and J.É. conceptualized the project. E.F. designed and performed most of the experiments including chemical syntheses, chromatographic method development, biotransformation in batch and in continuous-flow mode, collected and processed the analytical data (i.e., spectroscopic and chromatographic analyses), under the supervision of H.G. and É.J.; Z.S. performed and validated the NMR measurements; M.D. performed and validated the HR-MS measurements; P.S. performed the overexpression of the enzymes; D.V. and E.F. performed the affinity chromatography of *Cv*S-TA<sub>W60C</sub> at preparative scale. E.F., L.P., G.H., and J.É. analyzed and validated the experimental data. E.F. wrote the first manuscript draft, E.F. and L.P. prepared the figures and tables, E.F. and L.P. edited and wrote the final version of the paper.

## Funding

## Competing interests

E.F., Z.S., M.D., and J.É. are employees of Richter Gedeon Plc.—developer of the atypical antipsychotic drug cariprazine. The transaminase-based dynamic isomerization concept revealed by this study comprised the basis of a patent application related to industrial synthesis of cariprazine [Farkas, E.; Poppe, L.; Hornyánszky, G.; Incze, D.J.; Éles, J.; Sánta-Bell, E.; Molnár, Z.K.; Szemes, J.; Schneider, A.; Csuka, P. (Gedeon Richter Plc.-BME) Process to produce (1*r*,4*r*)-4-substituted cyclohexane-1-amines. *WIPO Int. Appl.* WO 2023/042081 A1 (2023)[55]. All other authors declare no competing interests.
