## [Peer Review File · Communications Chemistry]

Reviewers' comments:

Reviewer #1 (Remarks to the Author):

The manuscript "Transaminase-catalysis for trans-4-Substituted Cyclohexane-1-amines Involving the Key Intermediate of Cariprazine" describes a very interesting approach to 1,4 trans cyclohexanes. A few points are however not really made clear.

The authors write a clear introduction about amines and enzymes that can be utilized to prepare stereochemically pure amines and then nicely introduce the synthesis of cariprazine and their own target.

Then they describe the screening of the TAm they use Figure 2 is a little congested but shows that clearly only in one case trans selectivity is found while R and S enzymes are used. This is interesting but then the starting material is also a case that cannot simply be addressed with R or S selectivity, as the authors point out.

They then perform a "kinetic resolution" or is it a thermodynamic approach. Here it become very complex and the authors do not shed sufficient light on this. The process is given in Fig. 4. The final cis-trans ratio should be given by their thermodynamic equilibrium. In Fig. 3 the conversions Xtrans always stops within 5 h and is the stable. What is the driving force for the almost linear rate and then why does it stop completely? Does the initial pyruvate concentration play a role?

The authors have a very interesting system that they do not fully and clearly explain. This needs to be significantly improved.

Minor points:

In the text it is not clear how pure the enzymes are. Sequences should be provided. Also immobilisation should be mentioned in the text more clearly, not only in the experimental part.

In flow chemistry STY is a very important parameter, please include this.

In the flow chemistry also the residence times in the reactors should be provided.

Reviewer #2 (Remarks to the Author):

The manuscript "Transaminase-catalysis for trans-4-Substituted Cyclohexane-1-amines Involving the Key Intermediate of Cariprazine" reports an interesting and relevant biocatalytic synthesis of trans 4-substituted cyclohexan-1-amines catalyzed by transaminases. The manuscript describes a preliminary screening by using different transaminases in the format in the format of immobilized whole cells, and the application of the synthetic procedure in a fixed bed reactor using immobilized CvS-TAW60C. The results show a very well-done work regarding the obtention of a transaminase-catalyzed production of trans 4-substituted cyclohexane-1-amines from the corresponding cis/trans diastereomeric mixtures by selective deamination. A broad community could feel benefited and interested in the results once some points could be further clarified.

Major issues:

- In the so-called preliminary screening, different transaminases in the format of immobilized whole-cell catalysts are used. The authors report the amount of catalyst used (50 mg) but the information about

the quantity of expressed enzyme, the format of the immobilized-whole cell and the functional state of the enzyme is very scarce. Authors should describe and show a better deeper characterization of the catalysts used.

- In the experiments of the continuous flow reactor. The authors used an immobilized CvS-TAW60C. The data provided are not sufficient in terms of the amount of enzyme used and the functional state of the catalyst used.

- In the experiments of the continuous flow reactor, the authors describe the column characteristics in the methods section and the supplementary information. Data of flow rate are provided in the figure caption. However, data of residence time should be provided to allow a better and quicker understanding to the reader as well as data of reactor productivity and specific catalyst productivity. On the description of Figure 3 could be also convenient to mention that in the x-axis the operational time is used.

Other issues:

- The section “ Investigation of the diastereomer separation of cis/trans-1a-d by transaminase-catalyzed deamination” might be a bit confusing to readers. It does not provide any new results but a thought about further proceeding. Although the reflection described by the authors might interesting in an introduction context, the second paragraph contains too much introductory general information that does not match the features of the target reaction or reach of the evidences provided (e.g. comments regarding substrate of product inhibition).

- Authors operate the fixed-bed reactor up to 36 h (24 h under steady state) and claims about operational stability arise. Can the authors comment on the possibility of longer operations and long-term stability? Might be the observed stability an apparent one motivated by an excess of enzyme or too long residence times? Do the authors have evidence of the dependency of the conversion under steady state and the residence time?

- Have the authors considered kinetic studies to approach basic characteristic kinetic parameters of the reaction studied? This would be helpful to facilitate dissemination.

Reviewer #3 (Remarks to the Author):

The major claims of the paper:

-Identification of transaminases to catalyse the synthesis of four trans 4-substituted cyclohexanamines (with two pseudo asymmetric centres) in de >99%, including the chiral intermediate of the Cariprazine production route, by deamination of the diastereomeric amine mixture in continuous flow.

-In the cis-selective deamination step the authors observed that the yield of pure trans-isomers exceeded their original amount in the starting mixture and hypothesise that this could be explained by dynamic isomerization through the reversible biotransformation and consequent enrichment of the desired thermodynamically favoured trans-product.

-The single transaminase-catalyzed process allows for the enhancement of the productivity of industrial cariprazine synthesis.

Are they novel and will they be of interest to others in the community and the wider field?

Yes

-Single-enzyme dynamic cis-to-trans-isomerization based from thermodynamic control which favours enrichment of the trans-amine is a novelty.

-Three of four compounds are novel substrates for the transaminases tested in this study.

-The amination of 4-substituted hexanones with several enzymes in the submitted paper were studied in this publication: Fiorati, A., Berglund, P., Humble, M. S., & Tessaro, D. (2020). Application of transaminases in a disperse system for the bioamination of hydrophobic substrates. *Advanced Synthesis & Catalysis*, 362(5), 1156-1166

Is the work convincing, and if not, what further evidence would be required to strengthen the conclusions?

Yes, but:

Further evidence to strengthen the conclusions would be to include a reference, evidence or a more thorough explanation for the thermodynamic control of the trans-amine enrichment. The cyclohexane 1,4-trans-configuration is obviously more thermodynamically favourable than a 1,4-cis-configuration, but authors also state that "the trans-amine formation is also thermodynamically favored over the ketone formation." For me as a reader to fully understand this, it would be very helpful with an explanation or a reference to support this claim.

Will the paper influence thinking in the field?

Yes. The state of the art dynamic kinetic resolution (DKR) with transaminases for production of chiral amino compounds is based on the use of two-enzyme systems with complementary stereoselectivities. Poppe and co-workers here display a single-enzyme dynamic cis-to-trans isomerisation, in which the biotransformation equilibrium over time produces an excess of the desired trans-product, through a thermodynamically favourable process.

Further questions and concerns about the paper:

A few suggestions for clarity:

-Clearly state that the cis/trans ratio of the deamination starting material arises from the non-enzymatic reductive amination reactions.

-In table 1 the titles X and Y could be replaced with titles "Conversion" and/or "Yield" for a readership to more readily understand the table.

-Is it necessary to include one decimal point in the yields in table 1?

-I find the word "degradation" in this sentence misleading. Could, for instance, be replaced with "conversion": "Transaminases were identified enabling the degradation of the cis-diastereomer of four selected cis/trans-amines"

Comments on the appropriateness and validity of any statistical analysis, as well the ability of a researcher to reproduce the work, given the level of detail provided.

The synthetic procedures and the compound analytical data in the supporting information are well documented and reproducible.

Accept with minor revisions.

Reviewers' comments:

Reviewer #1 (Remarks to the Author):

The manuscript "Transaminase-catalysis for trans-4-Substituted Cyclohexane-1-amines Involving the Key Intermediate of Cariprazine" describes a very interesting approach to 1,4 trans cyclohexanes. A few points are however not really made clear.

The authors write a clear introduction about amines and enzymes that can be utilized to prepare stereochemically pure amines and then nicely introduce the synthesis of cariprazine and their own target.

Then they describe the screening of the TAmS they use Figure 2 is a little congested but shows that clearly only in one case trans selectivity is found while R and S enzymes are used. This is interesting but then the starting material is also a case that cannot simply be addressed with R or S selectivity, as the authors point out.

Response: Since the stereogenic unit in *trans*-4-substituted cyclohexane-1-amines is not a single center of asymmetry resulting in two enantiomers (having identical heat of formation/Gibbs free-energy and no thermodynamic preference) but a larger part comprising two centers of pseudoasymmetry at positions 1 and 4 of a cyclohexane ring resulting in two diastereomers ((having different heat of formation/Gibbs free-energy and therefore a certain thermodynamic preference), it is not surprising that R/S preference had no relationship with cis/trans preference.

They then perform a "kinetic resolution" or is it a thermodynamic approach. Here it become very complex and the authors do not shed sufficient light on this.

Response: The process "kinetic resolution" is a catalyst-based enantiomer selective approach in which the practically irreversible nature of the catalytic step is essential for the high selectivity. When reversibility is allowed in KR, the selectivity drops and under full thermodynamic control, the products should be racemic (irrespective of the degree of kinetic selectivity).

At first, we wanted to exploit a similar approach and perform a "kinetic diastereomer separation" based on a transaminase (TA) catalyst-based diastereomer selective process. In this case, an almost irreversible process would allow the fast conversion of the preferred diastereomer to the corresponding ketone helping the recovery of the unreacted slower reacting amine from a mixture with the forming ketone in such amount which was present in the original mixture.

Since the TA-catalyzed processes are reversible, the diastereomer selective deamination process is not fully irreversible and the partial reversibility allows the prevalence of thermodynamic consequences. Since the *trans*-diastereomers are thermodynamically more favored, a process exploiting the combination of kinetic selectivity and thermodynamic equilibrium can surpass the limitation of the *trans*-diastereomer amount in the original mixture by the dynamic isomerization.

To describe the behavior of the real system involving the transaminase and the two amines, the corresponding ketone, the pyruvate and the two enantiomers of alanine, kinetic characterization of all possible transformations (back and forth, each multistep; needing more than dozen rate constants) and correct energetics of each component under the reaction conditions would be required to set up a complex differential equation system having only numerical solutions. These enormous efforts are certainly out of the scope of the present work.

To better describe the situation, a part of the above discussion is added to the revised MS at line 160 of the original MS. At later points of the revised MS, further discussions are also added (see next response).

The process is given in Fig. 4. The final *cis*-*trans* ratio should be given by their thermodynamic equilibrium. In Fig. 3 the conversions X_{trans} always stops within 5 h and is the stable. What is the driving force for the almost linear rate and then why does it stop completely? Does the initial pyruvate concentration play a role?

Response: In the process using TA in packed-bed reactor under continuous-flow conditions (Fig. 3) a balanced mixture of kinetic and thermodynamic components. In continuous-flow system, the first regime of outflow composition (until 5-6 h) represents non-stationary state after which the flow system reaches stationary state. Due to the large amount of catalyst in the packed-bed-reactor (high catalyst to substrate ratio), the kinetic selectivity elements are more pronounced in the flow system than thermodynamics. This is why almost diastereomerically pure *trans*-amines are forming in PBRs, but equilibration allows partial isomerization (thermodynamic component) and thereby higher amounts of *trans*-products than being present in the original mixture. To better explain the kinetic and thermodynamic factors and the role of pyruvate concentration the following text has been added to the revised MS: "The fact that using 1 molar equivalent pyruvate (100% of amine acceptor) resulted in only 17-60% ketone formation could be due to a mixture of kinetic and thermodynamic reasons. Thus, in the process using TA in packed-bed reactor under continuous-flow conditions both kinetic and thermodynamic components play role. Due to the large amount of catalyst, kinetic selectivity is more pronounced and almost diastereomerically pure *trans*-amines are forming but partial equilibration (thermodynamics) allows isomerization to the more stable isomer and thereby formation of *trans*-amines in higher amounts than present in the original mixture (Fig. 4). Usually, increasing pyruvate concentration increases the rate of *cis*-deamination (mostly influenced by kinetic factors) but on the other hand increases

the proportion of the 1-cyclohexanone-compound in the final mixture (mostly influenced by thermodynamic factors).⁶⁵” (Ref. 65: The published WIPO Application includes further TA-catalyzed reactions at different substoichiometric pyruvate amounts)

The authors have a very interesting system that they do not fully and clearly explain. This needs to be significantly improved.

Response: As indicated in the previous responses, significant parts of discussions have been added to the text of the revised MS helping the understanding of the situation.

Minor points:

In the text it is not clear how pure the enzymes are. Sequences should be provided.

Response: As indicated in Materials section of the revised MS, “Details on enzyme coding plasmids, transaminase sequences, and immobilization methods are described in Section 1.2, Section 3.2, and 3.3. of Supplementary information, respectively.”

Moreover, Section 3.3 in the revised Supplementary information on the immobilization of CvS-TA_{W60C} in macroporous polymer beads contains a Figure showing a picture SDS-PAGE to demonstrate the purity of the enzyme applied for immobilization.

The required sequences were reported in the original works. These information were added to the Supplementary Information as Section 1.2: “Original data on cloning, sequence and 3D-structure of the TAs in this work: *Arthrobacter citreus* mutant CNB05-01⁵¹ (*ArS*-TA)⁵² [Seq. ID 16 in Ref. S1]; from *Arthrobacter* sp. KNK168 (*ArR*-TA)⁵³ [Uniprot code: F7J696, PDB code: 3WWH⁵⁴; mutated variant of *Arthrobacter* sp. KNK168 (*ArR*-TA_{mut11})⁵⁵ [PDB code: 3WWJ^{Error! Bookmark not defined.} and 5FR9⁵⁶], *Aspergillus terreus* (*AtR*-TA)⁵⁷ [Uniprot code: Q0C8G1, PDB code: 4CE5⁵⁸], *Chromobacterium violaceum* (CvS-TA_{W60C})^{59,510} [Uniprot code: A0A1R0MXM9, PDB code: 6SNU], and *Vibrio fluvialis* (*VfS*-TA)⁵¹¹ [Uniprot code: F2XBU9, PDB codes: 3NUI, 4E3Q⁵¹²].

Plasmids encoding transaminase from *Arthrobacter citreus* (*ArS*-TA), from *Vibrio fluvialis* (*VfS*-TA), from *Aspergillus terreus* (*AtR*-TA), from *Arthrobacter* sp. (*ArR*-TA) and its mutated variant (*ArR*-TA_{mut11}) were a kind gift of Prof. Wolfgang Kroutil (University of Graz, Austria). The plasmid encoding the W60C mutant of ω -transaminase from *Chromobacterium violaceum* (CvS-TA_{W60C}) was a kind gift from Prof. Per Berglund (KTH Royal Institute of Technology, Sweden).”

Also immobilisation should be mentioned in the text more clearly, not only in the experimental part.

Response: The description has been extended to better describe the purity and form of immobilization for the different kinds of experiments. Many examples were already published in our patent application (Ref. 64: Farkas, E. et. al. (Gedeon Richter Plc.) Process to produce (1*r*,4*r*)-4-substituted cyclohexane-1-amines. *WIPO Pat. Appl.*, WO 2023/042081 A1 (2023).) with details and enzymes, their purity and mode of immobilization, as we stated in the manuscript: “The isomer separation strategy (Fig. 1d) based on the diastereomer selective deamination of the *cis*-amines *cis*-**1a-d** was performed with *VfS*-TA and CvS-TA_{W60C} mostly in batch mode using different forms—native soluble enzyme, immobilized whole-cells, and purified form immobilized on polymer resin.⁶⁴”

Nevertheless, essential parts of purity and immobilization methodology were added to the text to maintain the communications characteristics of the manuscript.

Text added to the section on whole-cell TAs: “For these experiments, recombinant *E. coli* whole-cells overexpressing one of the above TAs were immobilized together with hollow silica microspheres as support by entrapment in a sol-gel system. Details on expression of *ArR*-TA, *ArR*-TA_{mut}, *AtR*-TA, *ArS*-TA, *VfS*-TA, and CvS-TA_{W60C} and immobilization of TA-expressing whole-cells were published in our preceding work.^{Error! Bookmark not defined.} The sol-gel entrapment method combined the advantages of cell-adsorption on silica microspheres providing good mechanical properties of the biocatalyst with high immobilization yield (~100% of the cells were retained; ~0.9 g of dry TA biocatalyst could be produced from 1 g of wet cells) by the entrapping silica matrix. The immobilization could be scaled up from g scale to 10 g scale without any noticeable problem.^{Error! Bookmark not defined.}”

Text added to the section on imm CvS-TA_{W60C}: “The CvS-TA_{W60C} (purified by standard Ni-NTA method) was immobilized on bisepoxide-activated macroporous aminoalkyl resins. Previous experiments with pure CvS-TA_{W60C} (overexpressed in *E. coli* and purified by standard Ni-NTA method) showed that the best resin-immobilized TA form with high operational stability could be obtained by covalent attachment of the enzyme to glycerol diglycidyl ether activated ethylamine-functionalized mesoporous polymer.^{Error! Bookmark not defined.}”

In flow chemistry STY is a very important parameter, please include this.

Response: The space time yield data based on the gross reactor volumes could be determined unambiguously and were added to Table 1.

In the flow chemistry also the residence times in the reactors should be provided.

Response: As the newly added section 3.4.4.2 in the revised Supplementary Information indicates, we could measure the mass of a dry imm CvTA_{W60C}-filled column vs. buffer loaded imm CvTA_{W60C}-filled column to gain information of the real eluent content in a column. These results and the following discussion have been added to the revised version of Supplementary information:

“This residence time (τ_{app}) should be considered as apparent for such macroporous biocatalyst systems since the solvent-filled volume (V_{solv}) is the sum of volume of interparticle solvent content (in which the compound molecules are moving at the same linear rate as the solvent molecules) and the volume of solvent content in the pores of the particles (within the pores the substrate and product molecules are moving only by diffusion, thereby this part is flow-rate independent). Therefore, the residence time calculated by using the solvent-filled volume (V_{solv}) containing unknown fraction of flow-independent in-pore solvent volume for the macroporous beads should be lower than a non-porous particles-filled system with the same solvent-filled volume (V_{solv}).”

Reviewer #2 (Remarks to the Author):

The manuscript "Transaminase-catalysis for trans-4-Substituted Cyclohexane-1-amines Involving the Key Intermediate of Cariprazine" reports an interesting and relevant biocatalytic synthesis of trans 4-substituted cyclohexan-1-amines catalyzed by transaminases. The manuscript describes a preliminary screening by using different transaminases in the format of immobilized whole cells, and the application of the synthetic procedure in a fixed bed reactor using immobilized CvS-TAW60C. The results show a very well-done work regarding the obtention of a transaminase-catalyzed production of trans 4-substituted cyclohexane-1-amines from the corresponding cis/trans diastereomeric mixtures by selective deamination. A broad community could feel benefited and interested in the results once some points could be further clarified.

Response: Thank you for the positive comment. We feel that the issues to be further clarified could be resolved by the revision.

Major issues:

- In the so-called preliminary screening, different transaminases in the format of immobilized whole-cell catalysts are used. The authors report the amount of catalyst used (50 mg) but the information about the quantity of expressed enzyme, the format of the immobilized-whole cell and the functional state of the enzyme is very scarce. Authors should describe and show a better deeper characterization of the catalysts used.

Response: As Reviewer 1 also noticed this issue, we described the whole-cell immobilization in more detail.

Nevertheless, essential parts of purity and immobilization methodology were added to the text to maintain the communications characteristics of the manuscript.

Text added to the section on whole-cell TAs: "For these experiments, recombinant *E. coli* whole-cells overexpressing one of the above TAs were immobilized together with hollow silica microspheres as support by entrapment in a sol-gel system. Details on expression of *ArR*-TA, *ArR*-TA_{mut}, *AtR*-TA, *ArS*-TA, *VfS*-TA, and *CvS*-TA_{W60C} and immobilization of TA-expressing whole-cells were published in our preceding work. Further details can be found in Supplementary information Section 3.2. The sol-gel entrapment method combined the advantages of cell-adsorption on silica microspheres providing good mechanical properties of the biocatalyst with high immobilization yield (~100% of the cells were retained; ~0.9 g of dry TA biocatalyst could be produced from 1 g of wet cells) by the entrapping silica matrix. The immobilization could be scaled up from g scale to 10 g scale without any noticeable problem."

- In the experiments of the continuous flow reactor. The authors used an immobilized CvS-TAW60C. The data provided are not sufficient in terms of the amount of enzyme used and the functional state of the catalyst used.

Response: According to your and Reviewer 1's notes, we extended our manuscript.

Text added to the section on whole-cell TAs: "The *CvS*-TA_{W60C} (purified by standard Ni-NTA method) was immobilized on bisepoxide-activated macroporous aminoalkyl resins. Previous experiments with pure *CvS*-TA_{W60C} (overexpressed in *E. coli* and purified by standard Ni-NTA method) showed that the best resin-immobilized TA form with high operational stability could be obtained by covalent attachment of the enzyme to glycerol diglycidyl ether activated ethylamine-functionalized mesoporous polymer."

Moreover, as noted in our response to Reviewer 1, Supplementary information has been significantly extended with description of the immobilization methods and the purity of *CvS*-TA_{W60C} (including SDS-PAGE characterization of the purified *CvS*-TA_{W60C} as Supplementary Figure S6).

- In the experiments of the continuous flow reactor, the authors describe the column characteristics in the methods section and the supplementary information. Data of flow rate are provided in the figure caption. However, data of residence time should be provided to allow a better and quicker understanding to the reader as well as data of reactor productivity and specific catalyst productivity.

Response: As we responded to Reviewer 1 as well, we could easily measure the mass of a dry imm *CvTA*_{W60C}-filled column vs. buffer loaded imm *CvTA*_{W60C}-filled column to gain information of the real eluent content in a column. This result and the following discussion have been added to the revised version of Supplementary information.

However, in such macroporous biocatalyst systems this would result in mixed data on interparticles solvent content (applicable for usual residence time estimations valid for nonporous systems) plus in-pore solvent content (in the solvent fraction within the pores substrate and product moves only by diffusion, thereby this part is flow-rate independent). Due to these reasons, residence time in such macroporous particles catalysts system is hard to estimate and cannot be considered as unambiguous data.

As a response to Reviewer 1, the space time yield data based on the gross reactor volumes could be determined unambiguously and were added to Table 1.

On the description of Figure 3 could be also convenient to mention that in the x-axis the operational time is used.

Response: Caption to Figure 3 is extended as follows: "... (the molar fractions of *trans*-1a-d—also considering the forming ketone 2a-d—in the effluent are shown as a function of operational time)."

Other issues:

- The section "Investigation of the diastereomer separation of cis/trans-1a-d by transaminase-catalyzed deamination" might be a bit confusing to readers. It does not provide any new results but a thought about further proceeding. Although the reflection described by the authors might be interesting in an introduction context, the second paragraph contains too much introductory general information that does not match the features of the target reaction or reach of the evidences provided (e.g. comments regarding substrate of product inhibition).

Response: According to your note, the superfluous introductory "Investigation of the diastereomer separation of cis/trans-1a-d by transaminase-catalyzed deamination" section has been removed and a portion of it has been added to the introduction of the next paragraph.

- Authors operate the fixed-bed reactor up to 36 h (24 h under steady state) and claims about operational stability arise. Can the authors comment on the possibility of longer operations and long-term stability? Might be the observed stability an apparent one motivated by an excess of enzyme or too long residence times? Do the authors have evidence of the dependency of the conversion under steady state and the residence time?

Response: The original Supporting information already contained information on this issue. However, to better highlight this issue, we have added the following text to the revised MS: "According to our preliminary investigations (Supplementary information S3.4.4.3 and Supplementary Figure S13) the immobilized CvS-TA_{W60C} biocatalyst-filled columns lost a part of their activity after 24 h stationary operation, with ~40-50% residual activity in the 36-70 h period."

- Have the authors considered kinetic studies to approach basic characteristic kinetic parameters of the reaction studied? This would be helpful to facilitate dissemination.

Response: To describe the behavior of the real system involving the transaminase and the two amines, the corresponding ketone, the pyruvate and the two enantiomers of alanine, kinetic characterization of all possible transformations (back and forth, each multistep; needing more than dozen rate constants) and correct energetics of each component under the reaction conditions would be required to set up a complex differential equation system having only numerical solutions. These enormous efforts are certainly out of the scope of the present work.

To better describe the situation, a section discussing the situation has been added to the revised MS (at line 160 of the original MS).

Reviewer #3 (Remarks to the Author):

The major claims of the paper:

- Identification of transaminases to catalyse the synthesis of four trans 4-substituted cyclohexanamines (with two pseudo asymmetric centres) in de >99%, including the chiral intermediate of the Cariprazine production route, by deamination of the diastereomeric amine mixture in continuous flow.
- In the cis-selective deamination step the authors observed that the yield of pure trans-isomers exceeded their original amount in the starting mixture and hypothesise that this could be explained by dynamic isomerization through the reversible biotransformation and consequent enrichment of the desired thermodynamically favoured trans-product.
- The single transaminase-catalyzed process allows for the enhancement of the productivity of industrial cariprazine synthesis.

Response: Thank you for this clear summary of the content.

Are they novel and will they be of interest to others in the community and the wider field? Yes

- Single-enzyme dynamic cis-to-trans-isomerization based from thermodynamic control which favours enrichment of the trans-amine is a novelty.
- Three of four compounds are novel substrates for the transaminases tested in this study.
- The amination of 4-substituted hexanones with several enzymes in the submitted paper were studied in this publication: Fiorati, A., Berglund, P., Humble, M. S., & Tessaro, D. (2020). Application of transaminases in a disperse system for the bioamination of hydrophobic substrates. *Advanced Synthesis & Catalysis*, 362(5), 1156-1166

Response: The previous results on amination of the phenyl substituted ketone 2d have been added to the revised MS as follows: "The phenyl substituted ketone 2d was already investigated with CvS-TA, CvS-TA_{W60C} and in aminations performed in organic solvents indicating similar *cis*-stereopreference as our results in aqueous systems.⁶⁴"

Is the work convincing, and if not, what further evidence would be required to strengthen the conclusions? Yes, but:

Further evidence to strengthen the conclusions would be to include a reference, evidence or a more thorough explanation for the thermodynamic control of the trans-amine enrichment. The cyclohexane 1,4-trans-configuration is obviously more thermodynamically favourable than a 1,4-cis-configuration, but authors also state that "the trans-amine formation is also thermodynamically favored over the ketone formation." For me as a reader to fully understand this, it would be very helpful with an explanation or a reference to support this claim.

Response: We thank the referee pointing out this inaccuracy. We removed the incriminated sentence from the revised MS and added the following text to the revised MS instead: "The fact that using 1 molar equivalent pyruvate (100% of amine acceptor) resulted in only 17-60% ketone formation could be due to a mixture of kinetic and thermodynamic reasons. Thus, in the process using TA in packed-bed reactor under continuous-flow conditions both kinetic and thermodynamic components play role. Due to the large amount of catalyst, kinetic selectivity is more pronounced and almost diastereomerically pure *trans*-amines are forming but partial equilibration (thermodynamics) allows isomerization to the more stable isomer and thereby formation of *trans*-amines in higher amounts than present in the original mixture (Fig. 4). Increasing pyruvate concentration increases the rate of *cis*-deamination (mostly influenced by kinetic factors) but on the other hand increases the proportion of the 1-cyclohexanone-compound in the final mixture (mostly influenced by thermodynamic factors)."

Will the paper influence thinking in the field? Yes.

The state of the art dynamic kinetic resolution (DKR) with transaminases for production of chiral amino compounds is based on the use of two-enzyme systems with complementary stereoselectivities. Poppe and co-workers here display a single-enzyme dynamic cis-to-trans isomerisation, in which the biotransformation equilibrium over time produces an excess of the desired trans-product, through a thermodynamically favourable process.

Response: Thanks for the positive opinion.

Further questions and concerns about the paper:

A few suggestions for clarity:

- Clearly state that the cis/trans ratio of the deamination starting material arises from the non-enzymatic reductive amination reactions.

Response: To clarify this point, we extended the c section of the legends of Table 1 as follows: ", influenced by the chemical reductive amination from the starting ketones 2a-d".

- In table 1 the titles X and Y could be replaced with titles "Conversion" and/or "Yield" for a readership to more readily understand the table.

Response: Meaning of x is molar fraction, not conversion The meaning of this measure is clearly resolved in c section of the legend. Y is replaced by Yield for sake of clarity

- Is it necessary to include one decimal point in the yields in table 1?

Response: Molar fraction and yield date are changed to two digits accuracy in Table 1.

- I find the word "degradation" in this sentence misleading. Could, for instance, be replaced with "conversion": "Transaminases were identified enabling the degradation of the cis-diastereomer of four selected cis/trans-amines"

Response: According to your valuable comment, "degradation" has been replaced by "amine-to-ketone conversion" (or equivalent phrase) at three positions in the revised MS.

Comments on the appropriateness and validity of any statistical analysis, as well the ability of a researcher to reproduce the work, given the level of detail provided.

The synthetic procedures and the compound analytical data in the supporting information are well documented and reproducible.

Accept with minor revisions.

Response: Authors are thankful for the valuable comments and the positive opinion of the referee.

REVIEWERS' COMMENTS:

Reviewer #1 (Remarks to the Author):

The revised manuscript "Transaminase-catalysis for trans-4-Substituted Cyclohexane-1-amines Involving the Key Intermediate of Cariprazine" can now be accepted.

Reviewer #3 (Remarks to the Author):

All of my comments were addressed appropriately by the authors.